# Transcriptome Analysis and Functional Characterization Reveal That *Peclg* Gene Contributes to the Virulence of *Penicillium expansum* on Apple Fruits

**DOI:** 10.3390/foods12030479

**Published:** 2023-01-19

**Authors:** Jiayu Zhou, Weifeng Gong, Tingting Tu, Jiaqi Zhang, Xiaoshuang Xia, Luning Zhao, Xinghua Zhou, Yun Wang

**Affiliations:** 1School of Food and Biological Engineering, Jiangsu University, Zhenjiang 212013, China; 2Center of Analysis, Jiangsu University, Zhenjiang 212013, China

**Keywords:** apple, effector, pathogenicity, *Penicillium expansum*

## Abstract

*Penicillium expansum* is the causal agent of blue mold decay on apple fruits and is also known to be the major producer of patulin, a mycotoxin that represents serious hazard to human health. Several mechanisms have been suggested to explain the pathogenesis of *P. expansum* in host plants. Secreted effector proteins are vital for the pathogenicity of many fungal pathogens through manipulating their hosts for efficient colonization. In this study, we performed a RNA-Seq analysis followed by computational prediction of effector proteins from *P. expansum* during infection of the host apple fruits, and a total of 212 and 268 candidate effector protein genes were identified at 6 and 9 h after inoculation (hai), respectively. One of the candidate effector protein genes was identified as a concanavalin A-like lectin/glucanase (*Peclg*), which was dramatically induced during the pathogen–host interaction. Targeted knockout of *Peclg* resulted in significant reduction in conidial production and germination relative to the wild type. Further studies showed that in addition to salt stress, the mutant was much more sensitive to SDS and Congo red, suggesting a defect in cell wall integrity. Pathogenicity assays revealed that the Δ*Peclg* mutant showed significant decrease in virulence and infectious growth on apple fruits. All these results suggest that *Peclg* is required for fungal growth, stress response, and the virulence of *P. expansum*.

## 1. Introduction

*Penicillium expansum* is the major causal agent of blue mold decay, which is known to be one of the most common postharvest diseases of apple fruits. In addition to causing significant economic losses worldwide, *P. expansum* also produces harmful mycotoxins, such as patulin, which is detrimental to human health and is also a potential carcinogen [1,2]. Due to the lack of host-based resistance to blue mold disease in commercial apple cultivars, chemical fungicides remain the primary strategy in the control of postharvest decay. However, as a result of their widespread use, fungicide-resistant populations of this pathogen have emerged, and most chemicals now have reduced efficacy to control blue mold decay of apple fruit [3,4]. Hence, innovative approaches are needed to develop new and more effective control strategies. Exploring the virulence determinants and understanding the molecular mechanisms of pathogenicity of *P. expansum*, which has been greatly accelerated by the releasing of the genomic sequence of this fungus [5,6], might lead to new mitigation strategies for postharvest blue mold decay.

While some postharvest pathogenic fungi can infect host plants directly, others such as *P. expansum*, an opportunistic fungus, infects fruit through natural opening or wounds inflicted before, during, and after harvest [7,8]. To successfully infect and colonize host plants, numerous virulence strategies have evolved in fungal pathogen to suppress the plant innate immune response, manipulate host physiology to accommodate microbial invaders, and thereby promote infection and disease development [9,10]. Among these strategies, a broad spectrum of secreted virulence factors is at the front line of host–microbe interactions, and these play important roles in plant infection [11,12].

During the past decade, many efforts have been made to identify effectors that contribute to the virulence of *P. expansum* [13]. For example, a series of works by Levin et al. [14,15] revealed that the deletion of *Penlp*1 gene and subtilisin-related peptidase *Peprt* gene in *P. expansum* resulted in obvious changes in mycelia morphology and structure, reduction in conidia production, and declined virulence on apple fruits when compared to the wild type, suggesting their important roles in the pathogenicity of *P. expansum*. Most recently, Blistering1, a protein with DnaJ domain, was identified as virulence factor in *P. expansum* through a T-DNA mutagenesis screening by Jurick et al. [16], in which they found that knockout of this gene resulted in reduced patulin production, decreased lesion diameter on apples, and impaired protein secretion, especially the extracellular secretion of plant cell-wall-degrading enzymes. However, characterization of effector proteins in *P. expansum* and investigation of their potential functions in plant infection and disease development have not been accomplished.

With the advent of fungal genome sequencing, omics approaches based on experimental analysis or computational prediction have been widely utilized for high-throughput screening of effector candidates [17,18,19,20]. For instance, through RNA-Seq analysis combined with SignalP prediction, 1728 putative secreted effector protein genes were predicted in *Magnaporthe oryzae* at 12, 24, 36, and 48 h post inoculation on barley leaves. The *MoSVP* gene, which encodes a protein with a hydrophobic surface binding protein A domain, was selected, and target deletion of this gene resulted in a reduction in pathogenicity, suggesting that MoSVP is a novel virulence effector of *M. oryzae* [18]. Therefore, with the aim to identify virulent effectors on a large scale and better characterize the molecular basis for the virulence, we performed transcriptome profiling of *P. expansum* during plant infection followed by in silico prediction to screen secreted virulence effectors. More specifically, a concanavalin A-like lectin/glucanase (named *Peclg*) gene, one of the candidate effector protein genes up-regulated in *P. expansum* during infection of apple fruits, was functionally characterized by gene knockout and revealed that *Peclg* plays an important role in the virulence of *P. expansum*. 

## 2. Materials and Methods

### 2.1. Strains and Growth Conditions

*Penicillium expansum* (isolate 3.3703) was obtained from China General Microbiological Culture Collection Center and maintained on potato dextrose agar (PDA) medium at 25 °C. For the preparation of conidia, 10 mL sterile water was added to the one-week-old PDA cultures. Conidia were collected by scraping the plates followed by filtration to remove the mycelium and then re-suspended in distilled water to a concentration of 10^8^ conidia per milliliter.

### 2.2. Scanning Electron Microscopy Observation

Uniform wounds (5 × 5 mm, deep × wide) were made at the equator of apple fruits, and a total of 20 μL *P. expansum* suspension (10^6^ conidia/mL) was artificially inoculated into each wound [21]. Treated apples were then kept in plastic baskets and covered with film wraps to maintain relative humidity of 90% at storage temperature of 25 °C. At 0, 6, 9, 12, and 24 h after inoculation (hai), fruit disks were prepared by cutting from the edge of lesion. Samples were then fixed in 2.5% glutaraldehyde/0.1 M PBS (pH 7.0) at 4 °C overnight. After the fixation procedure, samples were washed with PBS for three times to remove the extra glutaraldehyde, followed by dehydration in increasing concentrations of ethanol solutions (25%, 50%, 70%, 95%, and 100%). Samples were mounted on specimen stubs, sputter-coated with gold under vacuum, and imaged with a Hitachi S-4800 Field emission scanning electron microscope (Hitachi Medical Corporation, Tokyo, Japan).

### 2.3. cDNA Library Construction, Sequencing, and Effector Protein Genes Prediction

Samples were collected as above described at 6 and 12 hai for RNA extraction. Total RNA was isolated using Trizol RNA Extraction Kit (SK1321). RNA quality was determined by agarose gel electrophoresis, and RNA quantity was measured using Qubit RNA Assay Kit in Qubit 2.0 Flurometer (Life Technologies, Carlsbad, CA, USA). RNA-Seq libraries were constructed using a VAHTSTM mRNA-seq V2 Library Prep Kit for Illumina (Vazyme Biotech, Nanjing, China) following the manufacturer’s protocol. The preprocessed RNA-Seq reads were then mapped to the reference *P. expansum* genome (https://www.ncbi.nlm.nih.gov/genome/11336?genome_assembly_id=212204, accessed on 19 May 2022) using TopHat program (v2.0.11). 

Transcript abundance was normalized as fragments per kb of transcript sequence per million mapped fragments (FPKM) using Cufflinks (version 2.2.1). Differentially expressed genes (DEGs) were identified using the Cuffdiff v2.2.1, and the threshold was set with fold change ≥ 2.0 and cut-off of false-discovery rate (FDR) *p*-value < 0.05. Afterwards, GO (gene ontology) enrichment analysis was conducted using Blast2GO program, and KEGG analysis was performed using the online KEGG Automatic Annotation Sever (KAAS, http://www.genome.jp/kegg/kaas/, accessed on 19 May 2022). In silico prediction of secreted protein genes was performed according to Ökmen et al. [22]. In brief, *P. expansum* secreted protein genes were predicted using the SignalP program (https://services.healthtech.dtu.dk/service.php?SignalP-5.0, accessed on 19 May 2022) to determine the presence of N-terminal signal peptide, and subsequently, TargetP (https://services.healthtech.dtu.dk/service.php?TargetP-2.0, accessed on 19 May 2022) and TMHMM software (https://dtu.biolib.com/DeepTMHMM, accessed on 19 May 2022) program were used to identify and remove proteins predicted to be located at the mitochondrion and membrane.

### 2.4. Quantitative Real-Time PCR

Total RNA was reverse-transcribed into cDNA in a volume of 20 μL using the First Strand Synthesis Kit (Thermo Fisher Scientific, Waltham, MA, USA), according to the manufacturer’s instructions. qRT-PCR was performed on a StepOne Plus thermocycler (Applied Biosystems, Foster, CA, USA) utilizing SYBR Green Fast qPCR Master Mix (Sangon Biotech, Shanghai, China). The PCR procedure was set as follows: 3 min at 95 °C followed by 45 cycles at 95 °C for 3 s, 60 °C for 30 s, and 72 °C for 30 s. The relative expression level was calculated using the 2^−ΔΔCt^ method, and the 18S rDNA gene was used as a constitutive control. Each sample contains three analytical replicates and two biological replicates (four *P. expansum*-inoculated apple fruits for each). Primers used in qRT-PCR are listed in Appendix A. 

### 2.5. Construction of Peclg Deletion Mutant

*Peclg* knockout mutant (Δ*Peclg*) was constructed according to the methods described by Zhang et al. [23], with some modifications. *P. expansum* genomic DNA was isolated using a Fungi Genomic DNA Isolation Kit (Sangon Biotech, Shanghai, China). The upstream and downstream regions at the target site of *Peclg* were amplified from genomic DNA of *P. expansum* using *Peclg*-up-F/*Peclg*-up-R and *Peclg*-down-F/*Peclg*-down-R primer pairs, respectively. Hygromycin B phosphotransferase (hyg) gene was amplified from a binary vector pCAMBIA-1300 using hyg-F/hyg-R primer pair. The three fragments were fused together to generate the deletion allele of *Peclg* by overlap PCR using the knockout primer pair *Peclg*-knock-F/*Peclg*-knock-R. The resulting deletion allele was then directly introduced into the *P. expansum* protoplasts using a polyethylene-glycol-mediated transformation. The transformants were cultivated on PDA medium containing 120 μg/mL hygromycin B, and plates were incubated for 3–7 days. Hygromycin-B-resistant colonies were picked and confirmed by PCR amplification using the specific primers *Peclg*-out-F/*Peclg*-out-R, which were designed to detect the deletion allele of *Peclg* gene in the transformants, and *Peclg*-in-F/*Peclg*-in-R primers, which was specific to *Peclg* gene. 

### 2.6. Radial Growth, Conidia Germination, and Chemical Sensitivity Assays

For radial growth assay, an appropriate amount of conidial suspension (10 μL, 10^7^ conidia/mL) of the wild type and deletion mutant strain was inoculated at the center of PDA plate, respectively. After incubation for 7 d at 25 °C, the mycelia colony diameter was measured, and conidia produced on the plate were harvested in sterile distilled water. Conidia were counted using a hemocytometer and converted to the number of conidia per square centimeter of colony.

Conidia germination assay was performed according to Sun et al. [24]. Briefly, 10 μL of conidial suspension (10^7^ conidia/mL) was inoculated in PDB liquid medium. Numbers of germinated conidia were counted microscopically after incubation for 18 h to assess germination rate of conidia. Conidia were considered to be germinated when the germ tube length was greater than the conidia diameter. 

For chemical sensitivity tests, the wild type and Δ*Peclg* mutant were cultured on PDA plate containing 1 M NaCl, 10 mM H_2_O_2_, 0.04% SDS and 300 μg/mL Congo red, respectively. The colony diameters were measured after incubation at 25 °C for 7 days, and the percentage of radial growth inhibitions was calculated. 

### 2.7. Patulin Content Determination

Patulin content was determined with high-performance liquid chromatography (HPLC) as described previously [25]. Briefly, 50 mL apple juice medium was inoculated with 1 mL of *P. expansum* conidial suspension (10^7^ conidia/mL) and incubated at 25 °C in the dark for 3 d. The supernatant was filtrated through filtered through a 0.45 µm filter, followed by extraction with the same volume of ethyl acetate for two times. The collected upper layers were cleaned with 1.5% sodium carbonate and evaporated to dryness using a rotavapor system. The dried residue was dissolved completely with 1 mL of 0.2 mol/L acetic acid, filtered through 0.22 μm membrane filter, and then used for patulin quantification. HPLC analysis was carried out with a Shimadzu SPD-M20A HPLC system (Shimadzu, Kyoto, Japan). Chromatographic separation was achieved using a column of Shim-pack VP-ODS C18 (250 × 4.6 mm, 5 μm) at room temperature. The modified mobile phase, H_2_O with acetonitrile 90:10 (*v*/*v*), was used with a flow rate of 1 mL/min and isocratic mode. The detection wavelength was set at 276 nm. Three replicates (four flasks for each) were used in this assay. 

### 2.8. Virulence Assay

Apple fruits were surface-sterilized in 1% sodium hypochlorite solution for 2 min, rinsed in sterile distilled water three times, and air-dried at room temperature. Then, uniform wounds (5 × 5 mm, deep × wide) were made with a sterile nail around the equator of apple fruits. Artificial infection was performed by inoculating 10 μL conidia suspension (10^7^ conidia/mL) of the Δ*Peclg* mutant or the wild-type strain of *P. expansum* in each wound, while control fruits were inoculated with 10 μL sterile distilled water. The treated apple fruits were then kept in the dark at the temperature of 25 °C in a relative humidity of 90% for 9 d. The development of the disease lesions during the storage was examined by measuring two perpendicular diameters. Four apple fruits constituted a single replicate, and each treatment was replicated three times.

### 2.9. Statistical Analysis 

Statistical analyses were performed by using SPSS 16.0 statistic software package (SPSS Inc., Chicago, IL, USA). All the data were expressed as means ± standard deviation (SD), and subjected to one-way ANOVA with Duncan’s multiple-range tests. Differences were considered statistically significant for *p* < 0.05. 

## 3. Results

### 3.1. Microscopy Observation of P. expansum during Plant Infection

Scanning electronic microscopy was used to observe the infection process of *P. expansum* in apple fruits. As shown in Figure 1, the conidia of the *P. expansum* were elliptical in shape with a diameter of 3~5 μm and swelled at 6 hai. It was already possible to notice the conidial germination and the penetration of elongated germ-tubes through the apple tissues at 9 hai. Meanwhile, a layer of mesh-like substances secreted by fungal cells could be seen at the surface of wounds, which might facilitate the penetration. After 12 and 24 hai, the pulp of apple tissues began to undergo degradation due to the increased mycelium quantity in this region. It has been well-documented that production of effector proteins is a survival strategy for pathogens to overcome or inactivate the host immune response, resulting in progression of pathogenesis particularly at the early stage of infection [26]. Therefore, based on these findings, samples were collected at 6 and 9 hai and subjected to transcriptome analysis of *P. expansum* during the infection progress with the aim to identify the candidate effector genes.

### 3.2. Transcriptome Analysis of P. expansum during Plant Infection

cDNA libraries from *P. expansum*-infected fruit tissues at 0, 6, and 9 hai were generated and obtained a total of 85.6 Gb of sequence data composed of 285,211,054 clean reads. From these, 0.62%, 13.85%, and 18.01% of the cleaned reads in 0, 6, and 9 hai samples, respectively, were mapped to the genome of *P. expansum* (Appendix A). RNA-Seq analysis yielded 3255 differentially expressed genes (DEGs) with fold change ≥2 at 6 hai, with 2462 up-regulated and 793 down-regulated, and 3824 DEGs at 9 hai, with 3046 upregulated and 778 down regulated (Figure 2a and Appendix A). To validate gene expression profile, eight DEGs were randomly selected for qRT-PCR analysis. As illustrated in Figure 2b, the expression patterns of these eight genes were generally consistent with the results obtained by RNA-Seq (r^2^ = 0.76), confirming the accuracy and reliability of the data from RNA-Seq. Further gene ontology (GO) and Kyoto Encyclopedia of Genes and Genomes (KEGG) enrichment analyses of *P. expansum* DEGs indicated that the enriched GO terms (Appendix A) and KEGG pathways (Appendix A) of the DEGs were similar to those previously reported [27,28].

Fungal effector genes are specifically expressed during the infection stage, and most of the effector proteins were secreted via the classical endoplasmic reticulum secretory pathway, which involves an N-terminal signal peptide [29]. Therefore, in silico prediction was used to identify effector candidate genes in the set of up-regulated genes during the infection of *P. expansum* on apple fruit, yielding 212 and 268 secreted effector protein genes at 6 and 9 hai, respectively (Appendix A). In addition to a number of unknown-function proteins, as expected, a significant portion of the secreted proteins with biological function was enzyme, mainly including the glycosyl hydrolase family, lipase, peptidases, oxidoreductases, and nuclease. 

It is worth noticing that a number of secreted protein genes were identified as cell-wall-degrading enzymes, including concanavalin A-like lectin/glucanases (PEX2_021580, PEX2_031290, and PEX2_080610), pectin lyase (PEX2_016450, PEX2_031470, and PEX2_107230), pectinesterase (PEX2_013190), and a number of glycoside hydrolase members (PEX2_000950, PEX2_003900, PEX2_016600, and PEX2_097510), and many efforts have been made on members of this family for their potential role as virulence factors [30,31,32]. In this study, a concanavalin A-like lectin/glucanase gene (PEX2_031290, denoted as *Peclg*) was dramatically induced during the pathogen–host interaction. Therefore, we conducted further research focusing the on *Peclg* gene. To characterize the function of *Peclg* in *P. expansum*, the *Peclg* knockout mutants (Δ*Peclg*) with a hygromycin-resistance gene instead of the entire *Peclg* gene were constructed by homologous recombination (Appendix A). 

### 3.3. Peclg Is Involved in Fungal Growth, Patulin Production, and Chemical Sensitivity

To reveal the possible roles of *Peclg* in fungal growth and development, the radial growth, conidial production, and germination of the Δ*Peclg* mutant and wild type were also analyzed. As presented in Figure 3a, the Δ*Peclg* mutant grew more rapidly than the wild type on PDA plate. After 7 d of growth, the colony diameter of the wild type was 4.0 cm, whereas the colony diameter of Δ*Peclg* mutant reached 4.9 cm. Meanwhile, visual assessments showed that the mycelia of Δ*Peclg* mutant appeared lighter in color and much looser. Moreover, the conidia production of the Δ*Peclg* mutant on PDA plate was 0.6 × 10^5^ conidia/mm^2^, which is significantly lower than 1.5 × 10^5^ conidia/mm^2^ of the wild-type strain (Figure 3b). After 10 h of incubation, about 59.7% of Δ*Peclg* conidia were germinated compared to 72.3% of the wild-type conidia, indicating a slight reduce in conidia germination rate (Figure 3c). When the patulin production abilities were evaluated, no significant difference was observed between the wild type and the Δ*Peclg* mutant (data not shown).

To further investigate whether the *Peclg* is required for the adaptation of *P. expansum* to chemical stresses, the Δ*Peclg* mutant and wild type were incubated on PDA plate amending with different chemical reagents. As illustrated in Figure 4, the Δ*Peclg* mutant showed more sensitive to osmotic (NaCl) and cell wall (SDS and Congo red) stresses compared to the wild type; however, no significant difference was observed among them when exposed to oxidative (H_2_O_2_) stress. All these results indicated that the *Peclg* gene might be involved in the response to osmotic stress and is required to maintain the cell wall integrity.

### 3.4. Peclg Plays Important Role in the Virulence of P. expansum

To explore whether the *Peclg* gene is required for the virulence of *P. expansum*, pathogenicity tests on apple fruits were performed for the wild-type and Δ*Peclg* strains. Conidial suspensions of the wild-type and mutant strains were artificially inoculated on apple fruits, and then, the lesion diameter was measured after inoculation. As shown in Figure 5a, both wild-type and Δ*Peclg* strains induced disease on apple fruits. However, the diameter of disease lesion caused by the Δ*Peclg* strain was obviously smaller compared to that caused by the wild-type *P. expansum*, indicating a 25.5% reduction in lesion diameter 7 days after inoculation (Figure 5b). 

## 4. Discussion

Fungal pathogen secreted vast arrays of effector proteins during infection to aid in host colonization. Hence, the identification of virulence-related effector protein genes and functional characterization of their roles are critical for understanding the molecular mechanisms behind the pathogen–host plant interaction. In the present study, transcriptomic analyses of *P. expansum* during infection of apple fruits were performed, and a total of 212 and 268 candidate effector genes were identified at 6 and 9 hai, respectively. One of the effector genes was identified as *Peclg*, which was dramatically induced during the pathogen–host interaction. Therefore, further characterization of *Peclg* was performed to explore its potential roles in the infection and development of *P. expansum* decay on apples.

Concanavalin A-like lectin/glucanase, which belongs to glycoside hydrolase family and exhibits carbohydrate binding and glucanase activity, is known to be involved in various biological processes, such as morphogenesis, cell wall growth and extension, and suppression or prevention of plant immune responses [33,34]. As reported in the present work, target deletion of *Peclg* in *P. expansum* resulted in increased mycelial growth and reduced conidial production and germination relative to the wild type. Similar to our results, decreased conidial production was observed in knockout studies of glucanase gene in *Cochliobolus carbonum* [35] and *Pyrenophora tritici-repentis* [36]; however, the deletion of glucanase gene in these fungi also resulted in reduced growth rate, which was inconsistent with the Δ*Peclg* mutant. These data clearly indicated an important role of *Peclg* in regulating fungal growth and development; moreover, this regulation might be a species-dependent process, and further studies are needed to investigate the potential underlying mechanisms related to this regulation.

The sensitivity of the Δ*Peclg* mutant to some chemicals was also evaluated, and the results revealed that the Δ*Peclg* mutant showed defects in cell wall integrity because it showed increased sensitivity to cell-wall-destabilizing agents. It is reasonable since glucan is the main component of the fungal cell wall, and glucanase as well as other glucosidases are known to be essential for cell wall glucan synthesis and modification [37]. In *Saccharomyces cerevisiae*, the function of an endoglucanase gene (*Scw*10) was characterized by gene knockout, and deletion of the corresponding gene caused additive sensitivity towards cell wall stresses, suggesting a role in cell wall assembly or maintenance [38]. Similar results were also reported in *Neurospora crassa*, as endo-1,6-β-D-glucanase (neg1) gene deletion mutant does not affect fungal morphology but displayed no obvious phenotypic changes, but Congo-red and SDS, which affect fungal cell walls or membranes, remarkably inhibited the hyphal growth of the mutant at a concentration that did not inhibit growth in the wild type [39].

The plant cell wall provides the first physical barrier to prevent pathogen infection, and pathogens secrete numerous cell-wall-degrading enzymes, such as glucanases, galacturonases, and pectin lyases, that might contribute to the evading plant innate immune recognition and enable the penetration and spread within the host plant tissues by directly participating in the degradation of plant cell wall components, thereby playing an important role in the mechanisms of virulence during the entire infection cycle of the pathogen [36,40,41]. For instance, Fu et al. [35] found that knockout of *GLU*1 gene in *Pyrenophora tritici-repentis*, the major causal agent of wheat tan spot, resulted in a significantly reduced virulence, with a reduction up to 37% in diseased leaf area. Further cytological analysis of the infection revealed that the mutant produced significantly lower numbers of germ tubes and appressoria than the wild-type strain on susceptible wheat leaves. Another well-studied example is PsXEG1, a secreted protein from *Phytophthora sojae*, which exhibits xyloglucanase and β-glucanase activity and acts as an important effector during *P. sojae* infection but also acts as a pathogen-associated molecular pattern in soybean, where it can trigger defense responses including cell death [42]. Very recently, Ökmen et al. [32] functionally characterized Erc1, a conserved effector of smut fungus *Ustilago maydis* with an organ-specific virulence function in maize leaves. Their work also found that Erc1 binds to host cell wall components, displays 1,3-β-glucanase activity, and prevents β-glucan-induced host defenses, which is required for cell-to-cell extension, specifically in bundle sheaths cells. Moreover, although some research reported that patulin might play an important but not essential role in the pathogenicity of *P. expansum* [6,43,44], no significant difference in patulin production was observed between the wild type and Δ*Peclg* in this work, indicating that the reduced virulence of Δ*Peclg* is not due to its patulin production ability.

## 5. Conclusions

In this study, using a combination of transcriptome analysis of *P. expansum* during infection of apple fruits and in silico prediction, a large number of effector gene candidates were identified. A high proportion of the identified effector candidates were cell-wall-degrading enzyme genes, including concanavalin A-like lectin/glucanases, pectin lyase, and pectinesterase. A novel effector gene candidate, *Peclg*, was selected for functional characterization, and the results revealed that *Peclg* plays important roles in conidial production, germination, and the responses to environmental stresses. Moreover, target knockout of the *Peclg* gene also resulted in reduced virulence of *P. expansum* in apple fruits. All these findings suggested that *Peclg* functions as an important determinant of fungal growth, development, and virulence in *P. expansum*.

## Figures and Tables

**Figure 1 foods-12-00479-f001:**
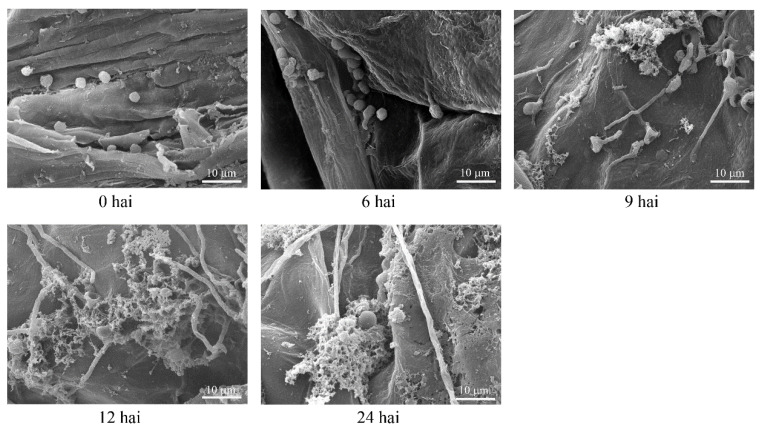
Scanning electron micrographs of *Penicillium expansum*-infected apple tissues at different time points.

**Figure 2 foods-12-00479-f002:**
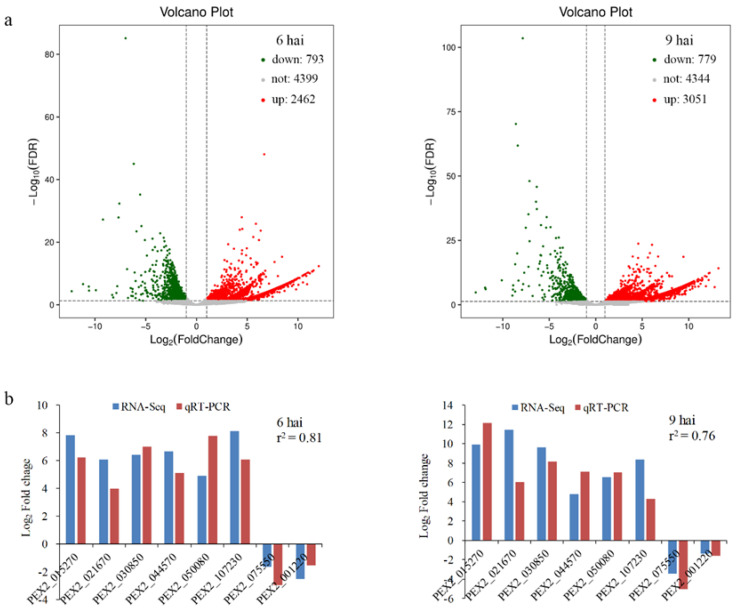
RNA-Seq analysis of *Penicillium expansum* during infection of apple fruits. (**a**) Volcano plot of all DEGs in *Penicillium expansum* at 6 and 9 hai. The red and green spots represent the up-regulated and down-regulated genes, respectively. (**b**) qRT-PCR validation of RNA-Seq data.

**Figure 3 foods-12-00479-f003:**
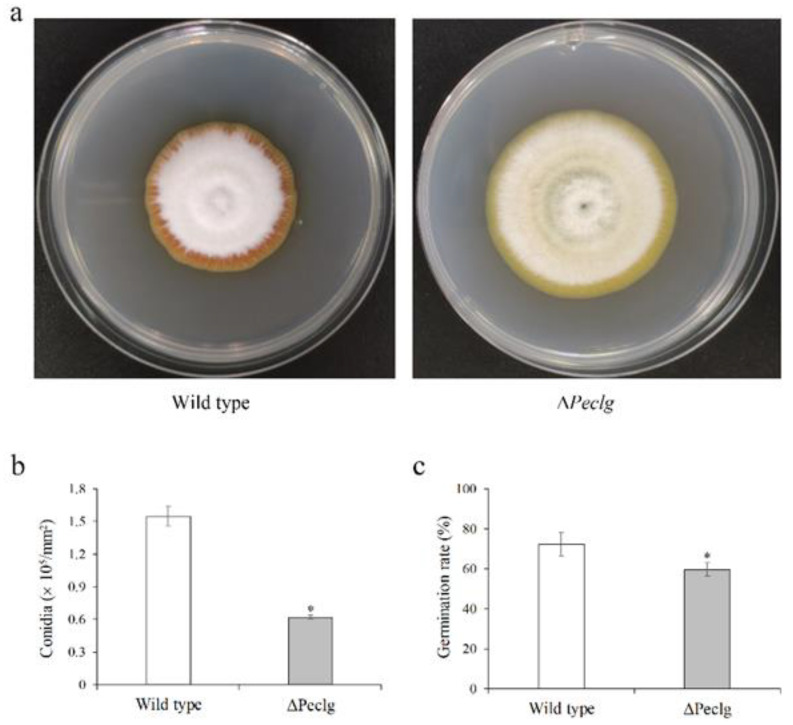
Effects of *Peclg* gene on (**a**) radial growth, (**b**) conidial production, and (**c**) germination of *Penicillium expansum*. Data are presented as means ± SD of three replicates. Asterisks indicate statistically significant differences (*p* < 0.05) between strains.

**Figure 4 foods-12-00479-f004:**
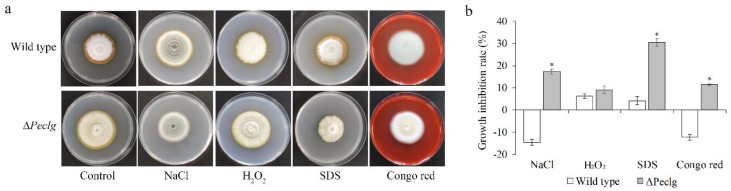
The *Peclg* gene regulates chemical stress responses in *Penicillium expansum*. (**a**) Colony morphology of the wild-type and Δ*Peclg* strains grown on PDA plate containing 1 M NaCl, 10 mM H_2_O_2_, 0.04% SDS, and 300 μg/mL Congo red. (**b**) Inhibition rate of growth in different stress treatments. The experiments were performed with three biological replicates. Data are presented as means ± SD of three replicates. Asterisks indicate statistically significant differences (*p* < 0.05) between strains.

**Figure 5 foods-12-00479-f005:**
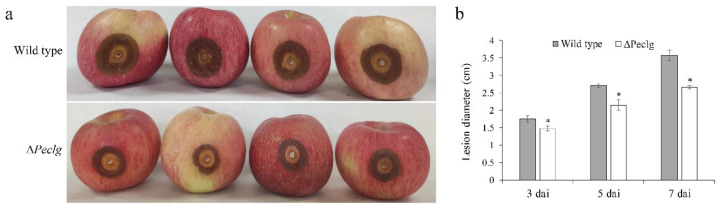
Blue mold lesions on apple fruits caused by the wild-type and Δ*Peclg* strain of *Penicillium expansum* 7 d after inoculation (dai) (**a**) and a graphical presentation of the lesion diameter at 3, 5, and 7 dai (**b**). Data are presented as means ± SD of three replicates, and each replicate contains four apples. Asterisks indicate statistically significant differences (*p* < 0.05) between strains.

## Data Availability

Data are contained within the article and Appendix A.

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
