# Peer review of "Transcriptome Analysis and Functional Characterization Reveal That Peclg Gene Contributes to the Virulence of Penicillium expansum on Apple Fruits"

_foods, 2023, doi:10.3390/foods12030479_

Round 1

Reviewer 1 Report

The aim of this study was identify virulent effectors large-scale and better characterize the molecular basis for the virulence for P. expansum during plant infection. The study design is acceptable. The study contains some valuable results and need suitable revisions.

Suggestions/ comments:

L3: Latin names (e.g. P. expansum) should be italic throughout the text.

L26-27: Citation row is missing.

L29: Again 'Penicillium expansum' should be italic and you should follow this throughout the text.

L44: You need more citation here not only the (7).

L57: et al. - change it throughout the text.

L67: 'Magnaporthe oryzae', again it should be in italic.

L80: Begining of the sentence should be Penicillium ...

L195: Figure should be placed after the text where it was first refered in the text.

L196: P. expansum. In the title of figure you should use a full name. Give also explanaition for hai in the title. 

Figs 2 and 3. P. expansum. In the title of figure you should use a full name.

Fig 2. Give also explanaition for hai in the title. 

Fig 4. H2O2 - give the right format

Fig 5. Give explanation for dpi in the title

L362: Conclusion section is missing.

References not fully follow the journal format. And again latin names in the title of cited works should be in italic.

Reviewer 2 Report

The manuscript is very well-written, scientifically sound and of significance in the food safety field. I have few comments:

1- Please write Peclg gene in the italic form in the title and throughout your manuscript

2- Please add the primers efficiency in the method section 2.4

3- Please add quality assurance to the methods sections 2.4, and 2.7

Reviewer 3 Report

Research article is well written and well presented. I have recorded some suggestion on various pages of the attached PDF, which are self-explanatory. 

Author Response

1. Research article is well written and well presented. I have recorded some suggestion on various pages of the attached PDF, which are self-explanatory.

Response: Thank you for your positive and constructive comments on our manuscript. We have revised the manuscript according to your suggestion. Please refer to the revised manuscript for details.
